

# Quantifying the local predictability of the 2021 sudden stratospheric warming event using a novel nonlinear method

Guiping Zhang[1], Xuan Li[2], Yang Li[1], Quanliang Chen[1], Xin Zhou[*1]

[1]School of Atmospheric Sciences, Chengdu University of Information Technology, Chengdu, 610225, China

[2]Ocean Institute of Northwestern Polytechnical University, Taicang, 215400, China

[*]*Corresponding to:* Xin Zhou (zhouxin18@cuit.edu.cn)





## Abstract

Sudden stratospheric warming (SSW) is identified as a key role in improving the winter
subseasonal-to-seasonal prediction, for its surface impacts up to two weeks through
stratosphere troposphere coupling. A better understanding of the predictability of the
SSW itself, thus, is fundamental. Most of the previous studies investigate the
predictability of SSW events using linear approaches and give the approximate
predictability.

In the study, we quantify the local predictability limit the 2021 SSW event, which
caused cold extremes across East Asia and North America, by applying a nonlinear
method, Backward Searching for the Initial Condition (BaSIC), within ERA5 reanalysis
data and the S2S reforecasts. The nonlinear method BaSIC is advanced because the
nature of SSW is a chaotic system with intrinsic properties, making it difficult to
measure its predictability with traditional linear methods. The local predictability limit
of this 2021 SSW event is estimated to be 17 days using BaSIC method, exceeding
previous estimations by one to two weeks using linear methods.

To gain further insight into where the errors may originate and propagate, we trace the
sources of forecast errors of this SSW to the area of the fastest error growth. At the
beginning of the SSW forecast, the overall forecast errors are relatively small over the
whole polar stratosphere; the errors grow slowly in the first 2 weeks, but increase
rapidly in the mid-high latitudes over central Eurasia (30°-60°E) and propagate into the
rest of Eurasia. This indicates that the forecast errors in the 2021 SSW event mainly
originate from the high altitude over central Eurasia.

## 1   Introduction

Sudden stratospheric warming (SSW) is an extreme event that occurs at 60-90°N near
the polar stratosphere at 10hPa. During a SSW event, the temperature of the polar
stratosphere can rise by 30–40 K within just a few days (Butler et al., 2017). SSWs are
classified based on the behavior of the circumpolar westerly winds: if the westerly
winds weaken but do not reverse, it is identified as a minor SSW; if they change to
easterly winds, it signifies a major SSW. Although major SSWs occur in both
hemispheres, the frequencies are low. Specifically, major SSWs occur about six times





per decade in the Northern Hemisphere (NH) (Charlton and Polvani, 2007). In contrast,
only one major SSW event has been recorded in the Southern Hemisphere (SH) that
occurred in September 2002 (Dowdy et al., 2004), which is relatively rare (Krüger et
al., 2005; Yamazaki et al., 2020; Van Loon et al., 1973). Circulation anomalies
generated by SSW events can be transmitted down into the troposphere and have
impacts on surface weather and climate, such as influencing the cold events (Baldwin
and Dunkerton, 1999; Deng et al., 2008; Quiroz, 2012). The mechanism producing
SSWs is explained by planetary waves theory in previous studies (Labitzke, 1981;
O'neill and Youngblut, 1982; Andrews et al., 1987), which posits that SSWs result from
tropospheric fluctuations propagating upward to the stratosphere and interacting with
the stratospheric flow (Matsuno, 1971). Some recent studies, however, suggest the
stratospheric state, instead of the tropospheric wave sources, can be the main cause of
the SSWs (Jucker, 2016; Birner and Albers, 2017; White et al., 2019; De La Cámara et
al., 2019). The uncertainties in the SSWs itself indicate a need to understand the SSWs
as a chaotic system with intrinsic properties.

Sudden stratospheric warming is on the S2S timescale that link the weather and climate
timescale, but SSWs are difficult to forecast due to diverse impact factors from both the
stratosphere and troposphere. In the 1970s, studies on the predictability of SSW events
have begun to emerge. Miyakoda et al. (1970) attempted to develop numerical
predictions of the breakdown of the circumpolar vortex in the winter stratosphere, but
their 14d GCM simulation for March 1965 failed to capture the occurred sudden
warming. The February 1979 SSW was the first successfully simulated as well as
observed SSW from space using newly introduced operational temperature soundings
of the stratosphere (Miller et al., 1980; Palmer, 1981b, a). Butchart et al. (1982)
successfully simulated the 1979 SSW 5 days before the warming peak though relied on
lower boundary conditions prescribed at the tropopause. Subsequent numerical
forecasts found that this SSW was predictable up to 5 and 10 days (Mechoso et al., 1985;
Simmons and Strüfing, 1983).

The forecast skill for SSW events in the last century was relatively low, but the forecast
lead time has been extended with a better understanding of SSW physics and an
improvements in numerical modelling by including relevant processes in the
stratosphere and mesosphere. Mukougawa and Hirooka (2004) found the 1998 SSW
can be predicted about 30 days before the onset date using JMA forecast system for fair



reproduction of the wave-flow interactions that facilitate the generation of WN 1 planetary waves. Tripathi et al. (2015) estimated the predictability of 2013 SSW using multi-model NWP systems. They noted that the SSW could be predicted by all models at 10 days in advance, while only some models predicted weakening of westerly wind at 15 days in advance. Taguchi (2016) analyzed the predictability of SSWs from 1979 to 2012, noting that the predictability depends on the geometry of the polar vortex, with certain configurations being more predictable. Rao et al. (2021) examined the predictability of the 2021 SSW using 9 models from the S2S database and found that the SSW is predictable for no more than two weeks in advance, largely due to adverse tropical forcing. It is now generally accepted that SSWs can be predicted with certainty 1 to 2 weeks in advance (Domeisen et al., 2020).

There are many challenges in quantifying the predictability of SSW events. Taguchi (2018) analyzed the predictability of SSWs from 1979 to 2012 and suggested that the predictability of SSWs depends on the geometry of the polar vortex. Generally, vortex splitting events are more difficult to forecast than vortex displacement events (Taguchi, 2018; Song et al., 2020). Higher forecast skills are also likely associated with models that have an enhanced representation of stratosphere (Allen et al., 2006; Marshall and Scaife, 2010). In addition, the forecast skill of a single model for different SSWs varies due to the different phenomenology and generation mechanisms of vortex splitting and vortex displacement events (Karpechko, 2018; Rao et al., 2019). Additionally, the predictability of troposphere is susceptible to model biases affecting planetary waves and depends on their propagation timescale (Noguchi et al., 2016). In addition, the predictability of stratosphere is likely to be influenced by the state of the stratospheric background prior to an SSW (De La Cámara et al., 2017). The quantification of the SSW predictability, a chaotic system in nature, remains an open question.

In this study, we use a nonlinear approach for this task, to offer a new perspective in the up limit of the SSW predictability. The importance and advantages of using a nonlinear method in investigating atmospheric predictability have been documented by previous studies, but this is the first time that this nonlinear method to be used in estimating the SSW predictability. After Lorenz (1963) introduced the linear singular vector (SV), various linear methods have been developed to estimate predictability, including the signal-to-noise ratio (SNR), the anomaly correlation coefficient (ACC), the Lyapunov exponent (LE), and the local LE (LLE). However, due to the chaotic nature of





atmosphere, using the linear methods to study the atmospheric predictability has some limitations. They are unable to capture the error dynamics in the nonlinear regime (Nese, 1989; Yoden and Nomura, 1993). The nonlinear method was then developed to overcome this problem, including the nonlinear LLE (NLLE) and the backward NLLE (BNLLE) (Ding and Li, 2007; Ding et al., 2008; Li et al., 2018). Yet both NLLE and

BNLLE methods might be susceptible to uncertainties within the forecast models (Li et al., 2018). Li et al. (2023b) developed the backward searching for the initial condition (BaSIC) method to quantify the predictability of extreme events, which will be used in this study. The method utilizes a dynamical indicator that reflects the essential characteristics of predictability, avoiding the influence of numerical model drift errors,

which suggests it is a stable and efficient approach. It has been applied in estimating the predictability in extreme cold events and has been proved to be an effective approach in case of extreme events (Li et al., 2023b). The method is appliable to the major SSW, which itself can be considered as an extreme event for the stratospheric temperature can reach more than 70 K relative to the long-term mean.

In early January 2021, a major SSW occurred in the NH and caused striking cold extremes across East Asia and North America, which had caused recording low temperatures in both China and South of North America and notably 151 deaths in Texas during the mid-winter of 2020/21 (Zhang et al., 2021). During the January 2021 SSW, Easterly winds persisted for more than 20 days, and the stratospheric polar vortex

split into two centers at the beginning of January (Rao et al., 2021). Rao et al. (2021) and Cho et al. (2023) investigated its predictability using linear methods, determining it as two weeks in rough. In contrast, we study it using the BaSIC nonlinear method to quantitatively estimate the predictability limit of this event. We further trace the dynamical growths of forecast error.

The remainder of this paper is organized as follows. Section 2 briefly introduces the datasets, the fifth generation European Center for Medium Range Forecasting Reanalysis (ERA5; Hersbach et al., 2020) reanalysis and S2S reforecasts of European Centre for Medium-Range Weather Forecasts (ECMWF) and China Meteorological Administration (CMA), and BaSIC methodology. Section 3 studies the local

predictability of the 2021 SSW event and the sources of forecast errors. Finally, discussion and conclusion are presented in Section 4.





## 2 Data and methods

### 2.1 Datasets

This study used the ERA5 reanalysis data and the ECMWF and CMA models from the S2S prediction database (Vitart et al., 2017) to study the local predictability of the 2021 major SSW in the NH. The daily averages of ERA5 reanalysis data were used as the validation data, with a horizontal resolution of $0.5° \times 0.5°$. Operational subseasonal hindcasts from two numerical weather prediction centers are employed in the study, the

CMA and the ECMWF. The CMA hindcasts are initialized twice a week (Monday and Thursday), and its forecast length is up to 60 days. There is total four ensemble members, including one control member and three perturbed members. Same as the CMA hindcasts, ECMWF hindcasts are also initialized twice a week (Monday and Thursday). Differently, the ECMWF hindcasts have 11 ensemble members, including one control

and ten perturbed members. Mode details of CMA and ECMWF S2S hindcasts can be seen online (https://confluence.ecmwf.int/display/S2S/Models). Besides, the two S2S hindcasts have the same horizontal resolution of $0.5° \times 0.5°$ as the reanalysis data. To investigate the local predictability of the 2021 major SSW in the NH, the selected study period is from November 2020 to March 2021.

**2.2 Backward Searching for the Initial Condition (BaSIC) method**

    Attractor radius (AR) is the intrinsic metric of chaotic system, and it represents the average distance between the mean state and all other states on a compact attractor (Li et al., 2018). The AR indicates the predictability, and can be used to effectively quantify the predictability limits of chaotic system nonlinearly (see Appendix for more

information about the AR method). Based on the rationale, a novel method BaSIC, is developed based on the AR method in the work of Li et al. (2023b). Once the ES is identified, we need to work backwards to find the CIS, and subsequently obtain the upper predictability limit. The BaSIC method is applied to quantify the predictability of extreme events (Li et al., 2023b), and that's why we use this method instead of the

AR method in the study.





In a phase space, a random state $x_0$ on a dynamical trajectory has a local predictability limit. And if $x_0$ loses its predictability at the state $x_1$, then the time interval between the two states is the local predictability limit of $x_0$. Conversely, an arbitrary state $x_1$ has a CIS $x_0$ that can forecast the given state $x_1$. Thus, the timespan between the two states is the maximum prediction lead time (MPLT) of the state $x_1$. Based on this rationale, a given ES has a CIS, and the MPLT of the ES is the timespan between the ES and its CIS. That is, the CIS must be determined first to obtain the MPLT.

If there is time series X ($[x_1, x_2, ..., x_{ex}, ...]$), and $x_{ex}$ is the ES. Based on the above rationale, the CIS of the ES $x_{ex}$ should be determined first. We know that every ES has a CIS. Therefore, we just search the CIS backward. From equation (8), a given state $x_0$ loses its local practical predictability when the $\bar{e}(x_0, \delta_0, t)$ exceeds the AR. It means the state $x_0$ is the CIS. Thus,

$$J(x_0^*) = 0, \tag{1}$$

where

$$J(x) = \| \bar{e}(x, \delta, t) - AR \|, \tag{2}$$

$x$ denotes the state that precedes the ES $x_{ex}$, and δ represents the error perturbed on the state $x$. The CIS $x_0^*$ of the ES can be obtained by solving equations (9) and (10), and the MPLT of the ES $x_{ex}$ is the timespan between the ES $x_{ex}$ and the CIS $x_0^*$. Since the AR is the threshold in equation (10), the MPLT indicates the local practical predictability of the ES $x_{ex}$ (Li et al., 2023).

## 3 Results

Figure 1 compare the polar temperature and zonal wind of the 2021 SSW event with other ten historical SSW events since 2003. The polar temperature increased rapidly in early January, peaking at approximately -30°C, 20°C higher than climatology on 5 January, and readily became an outlier. Likewise, the zonal westerly wind decreased throughout December, and shifted to easterly on 5 January. Despite a zonal westerly shift between 21 January and 1 February, it was deemed a single SSW event (Rao et al., 2021). This 2021 SSW event is a major event, and its onset date is 5 January. Known for its split nature, it presents increased forecast challenges (Taguchi, 2018; Rao et al., 2021). We will first assess the performance of the ECMWF and CMA models for the

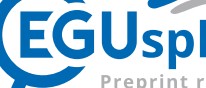



2020/21 winter and analyze their forecast skill to obtain a rough range of the predictability of this event before the quantification using the BaSIC method.

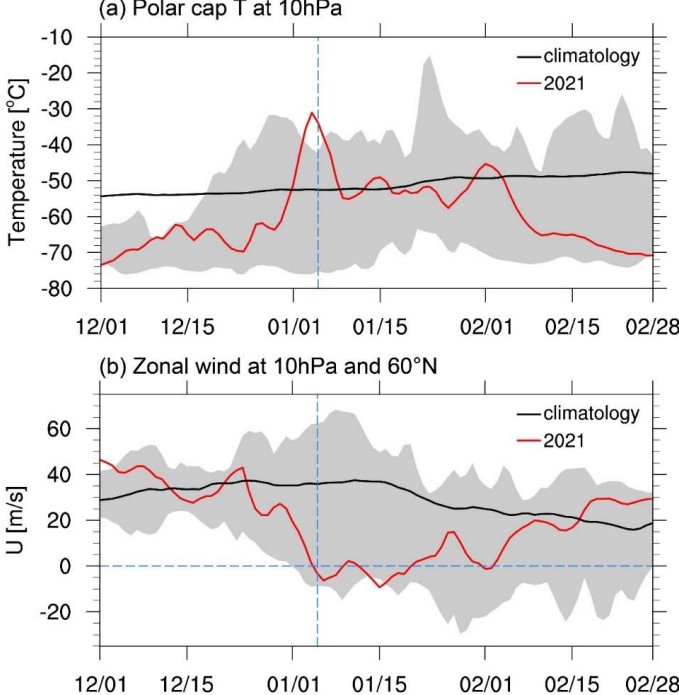

Fig.1 (a) Average polar temperature and (b) zonal mean zonal wind for 60°N at 10hPa in the polar region from December 2020 to January 2021. The solid red line indicates polar temperature and zonal wind, respectively for the 2021 SSW, and black represents the climatology. The shaded area indicates the range of occurrence of 10 major SSWs from 2003-2019 (https://csl.noaa.gov/groups/csl8/sswcompendium/majorevents.html) using ERA5 data. The vertical blue dashed line denotes the onset date of the SSW.

We first evaluate the ability of the two models, ECMWF and CMA, in forecasting zonal winds in winter (From December 2020 to January 2021). Figure 2 displays the average forecast errors of zonal winds at 60-90°N with the forecast time of 0-15 days. Forecast errors are small initially, and grow as the forecast time increases (Fig2 a, e). For the ECMWF, at the forecast time of 5 days, the positive forecast errors are mainly located over Northern America, while the negative forecast errors are mainly lied over Eurasia.



As the forecast time (10 and 15 days) further evolves, the range and magnitude of forecast errors have grown. The CMA shows larger forecast errors than those of ECMWF. At the forecast time of 5 days, most region have positive forecast errors up to 6 m/s. At the forecast time of 10 and 15 days, differently, a large areas of negative forecast errors are also existed. The negative forecast errors mainly spanned over eastern Eurasia and south Greenland. Overall, the ECMWF has smaller average forecast errors than those of CMA, indicating a better performance during this winter.


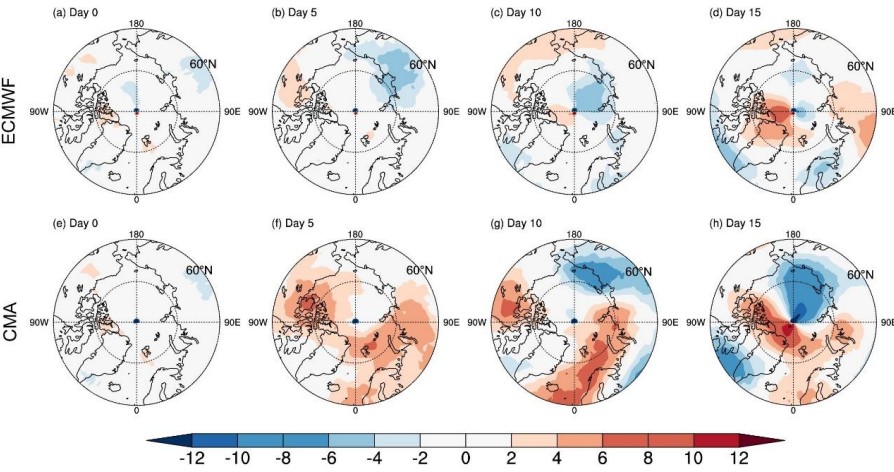

Fig. 2 Spatial distribution of forecast errors of zonal wind (north of 60°N) of forecasting 0-15 days by ECMWF and CMA centers from December 2020 to January 2021.

We also compared the RMSE and spread of zonal wind during the winter. For the ECMWF and CMA, they have the same RMSE at the initial time (about 3 m/s). As the forecast time increases, RMSEs both grow. However, the RMSE of CMA consistently remains larger than that of ECMWF. It is consistent with forecast errors of the two centers (shown in Fig. 2). The spread of ECMWF and CMA have the same tendencies
as the RMSE. Differently, the CMA spread is always lower than the ECMWF spread. It is mainly related to the different ensemble forecast schemes and ensemble members.





Generally, a higher spread correlates with better forecast skill, further suggesting that ECMWF has a better performance over CMA.

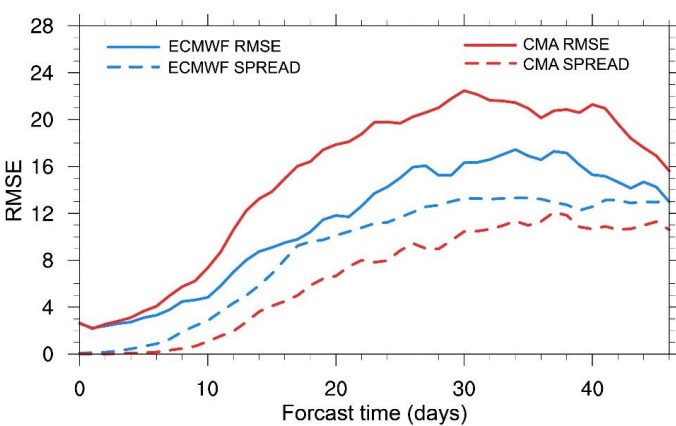


Fig. 3 RMSE and spread of zonal mean zonal wind as a function of forecast time from ECMWF and CMA centers. The solid and dashed lines represent RMSE and spread, respectively, while the blue and red indicate ECMWF and CMA, respectively.

We now analyze the forecast skill of the 2021 SSW event using the two models. Figure 4 shows the ERA5 reanalysis and the hindcasts initialized at different lead weeks. For the ECMWF, when the initialized dates are one or two weeks prior to the onset date of the 2021 SSW event, the ERA5 reanalysis is included in the range of forecasts for all ensemble members and very close to the ensemble mean (Figs. 4a, b). Yet for the CMA,

the forecasts and the reanalysis have slight differences when initialized dates is two weeks in advance (Fig. 4e). When initialized dates are three weeks prior to onset of the 2021 SSW event, the forecasts and the reanalysis show larger differences compared to one or two weeks in advance. Thereafter, the forecasts are unable to accurately capture the observed conditions represented by the ERA5 reanalysis From Figs. 4c and f, all

the forecast members are higher than the reanalysis. It indicates that the predictability of the 2021 SSW event is less than three weeks.



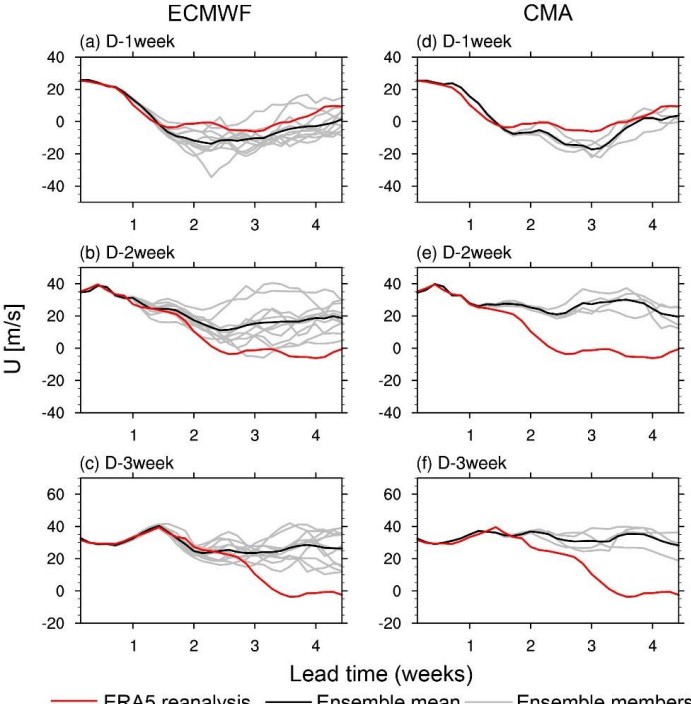

Fig. 4 The ERA5 analysis, and the hindcasts of ECMWF (a ~ c) and CMA (d ~ f) initialized (a,

d) one week before, (b, e) two weeks before, and (c, f) three weeks before the 2021 SSW onset

date. The red solid line shows the ERA5 reanalysis, the black solid line shows the ensemble

mean, and the gray solid line shows the ensemble member.

Figure 5 shows forecast errors of the onset (5 January 2021) of this SSW event

initialized at different lead weeks using the ECMWF and CMA models. When the

initialized date is one week in advance, both ECMWF and CMA show small forecast

errors for the 2021 SSW event onset. Large positive forecast errors are mainly located

at mid-high latitudes of the northern North Atlantic, whereas negative forecast errors

are spanned at mid-high latitudes of the Eurasia. When the initialized date is two weeks

in advance, the forecast errors grow obviously in both models. Both ECMWF and CMA

show mainly positive forecast errors in the high latitudes (60°N -90°N). The forecast

errors are generally distributed around the entire polar circle and are higher for CMA

than for ECMWF. When the initialized date is three weeks in advance, the forecast



errors become larger across the high latitudes, which suggests that both ECMWF and

CMA have lost their forecast skills. And it is consistent with the result in Figs. 4c and

f.

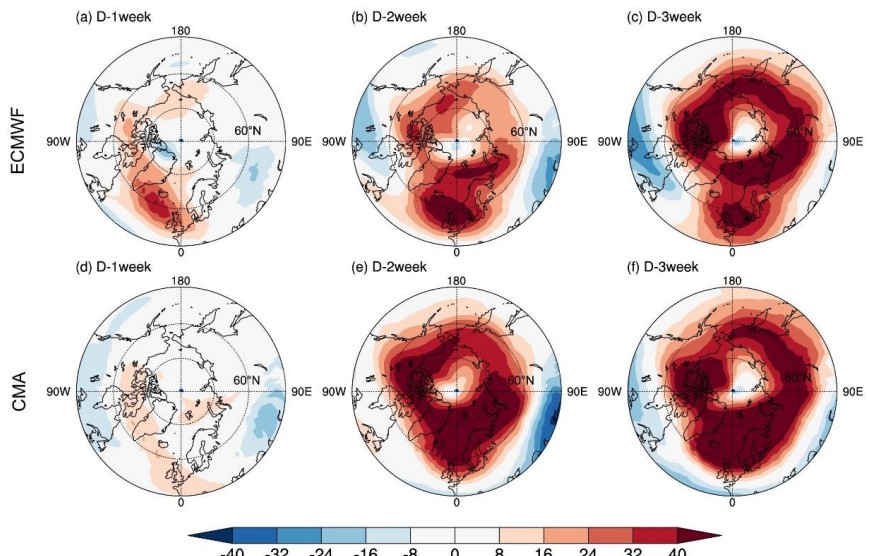

Fig.5 Spatial distribution of forecast errors (north of 45°N) in the ECMWF (a ~ c) and CMA (d

~ f) hindcasts initialized (a, d) one week before, (b, e) two weeks before, and (c, f) three weeks

before the 2021 SSW onset date.

We now find that the predictability of the 2021 SSW event is less than three weeks, and

the ECMWF performs better than CMA. Next, we will quantify the local predictability

of the 2021 SSW event. The BaSIC method is employed to quantify the upper limit of

local predictability of the 2021 SSW event. For the BaSIC method, two dynamical

indices, the AR and the RMSE, are the two terms that determine the estimation. A

lareger AR means higher predictability; while larger RMSE means greater forecast

errors between the forecasts and reanalysis. Figure 6a presents the AR of zonal winds

at 10hPa, derived from the ERA5 reanalysis dataset. The AR has a regular circular

latitude distribution in the NH. That is, on the same latitude circle, the AR values are





almost the same.The AR is largest near the equator while in the mid-low latitudes, the
AR is the smallest. At high latitudes (north of 60°N), the AR remains large, second to
that of the equator. Spatial patterns of the AR are dependent on dynamical instabilities
(Zhao et al., 2021; Li et al., 2023a). Higher variability resulting from the baroclinic
instability leads to a larger AR in the middle latitudes, wheras barotropic and convective
instabilities in the low latitudes result in a smaller AR. Li et al. (2023a) caculated the
AR of 2m temperature over East Asia and found smaller AR at low latitudes, with larger
AR at middle and high latitudes. Zhao et al. (2021) analyzed the AR of the geopotential
height at various pressure levels and pointed that the AR is the lowest over the tropics
and highest over middle latitudes in the SH. Fig. 6b shows the AR on the 60°N latitude
circle. It can be noticed that the AR has a small fluctuation at each longitude on the
60°N latitude circle, so we use the average AR to represent the AR for the whole latitude
circle. And the average AR is 13.70 m/s. Next, we will use the the average AR as the
AR mentioned in the BaSIC method.

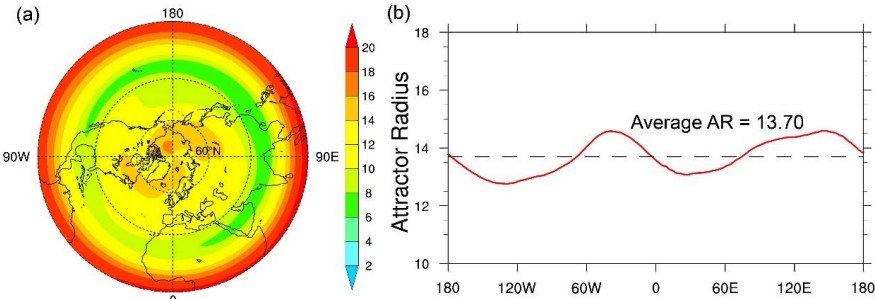

Fig. 6 Attractor radius (AR) of (a) zonal winds at 10hPa in the NH and (b) zonal mean zonal
winds at 60°N. The dashed line indicates the average AR at 60°N, which is about 13.70 m/s.

According to the BaSIC method, we need to determine the ES and CIS states of this
SSW event so the timespan between the two states is the upper limit of its predictability
(see Method for more information). In this study, the zonal wind of the 2021 SSW event
oneset is the ES state . Figure 7 presents the spatial structure of the zonal wind of the
2021 SSW event onset. In the middel latitudes, the westerly winds previal south of 60°N



while easterly winds dominate north of 60°N. On the latitude circle at 60°N, the zonal westerly winds have shifted to easterly winds in some regions. And in some other regions, the zonal westerly winds are very weak and show a tendency to transition to easterly winds.


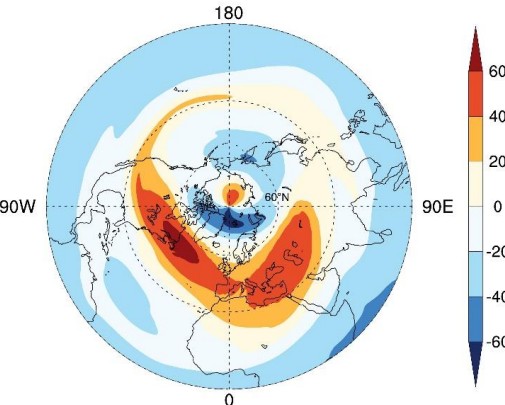

Fig. 7 Spatial structure of the zonal wind in the NH of the onset of the 2021 SSW event.

We then search the CIS and obtain the timespan between the ES and CIS. To eliminate

the effect of random noise, we perform a five-point moving average of the RMSEs. Considering that the ECMWF and CMA hindcasts are initialized twice a week (Monday and Thursday), the mean of hindcasts initialized from Monday and Thursday is represented to the average hindcast of this week. Using the hindcasts and ERA5 reanalysis, the RMSE can be calculated. Figure 8 shows the variations of RMSEs as the

function of the forecast lead time for the ECMWF and CMA. For the ECMWF, when the initialized date starts a week in advance (average hindcast of 26 and 29 December), it takes two weeks for the RMSE to exceed the AR (fig. 8a), demonstrating that the local predictability limit is two weeks. Therefore, the zonal wind of 2021 SSW onset can be well forecasted, when the initialized date is 26 or 29 December. It also signifies

the local predictability of the zonal wind of 2021 SSW onset is longer than one week. For the CMA, the RMSE takes three weeks to reach the AR (fig. 8d). Same as the ECMWF, the zonal wind of 2021 SSW onset can be well forecasted, when the initialized



date is 26 or 29 December. Therefore, it further proves that the local predictability of the zonal wind of 2021 SSW onset is longer than one week.

When the initialized date starts two weeks in advance (average hindcast of 19 and 22 December), it takes less than two weeks for the RMSE of CMA model to exceed the AR (fig. 8e), indicating the failure to forecasting the 2021 SSW onset. However, for the ECMWF, the RMSE just uses two weeks to reach the AR (fig. 8b), demonstrating that the 2021 SSW onset can be well forecasted from 19 or 22 December 2020. Whether

earlier initialization dates can still forecast the 2021 SSW onset? We further compared the hindcasts and ERA5 reanalysis starting from earlier dates. Figure 8c and f show the variations of the RMSE initialized from three weeks in advance. For the ECMWF, the RMSE takes about 17 days to exceed the AR. The SSW onset can't be forecasted successfully when the initialization begins either on 12 or 15 December. It is the same

for the CMA. Hence, the local upper predictability of SSW onset is no longer than three weeks. And the earliest initialization date which can well forecast the 2021 SSW onset is 19 December 2020. According to the BaSIC method, the zonal wind condition on 19 December 2020 is the CIS of the 2021 SSW onset. Therefore, the upper predictability limit of the 2021 SSW onset is 17 days (the time span between 19 December of 2020

and 5 January of 2021).



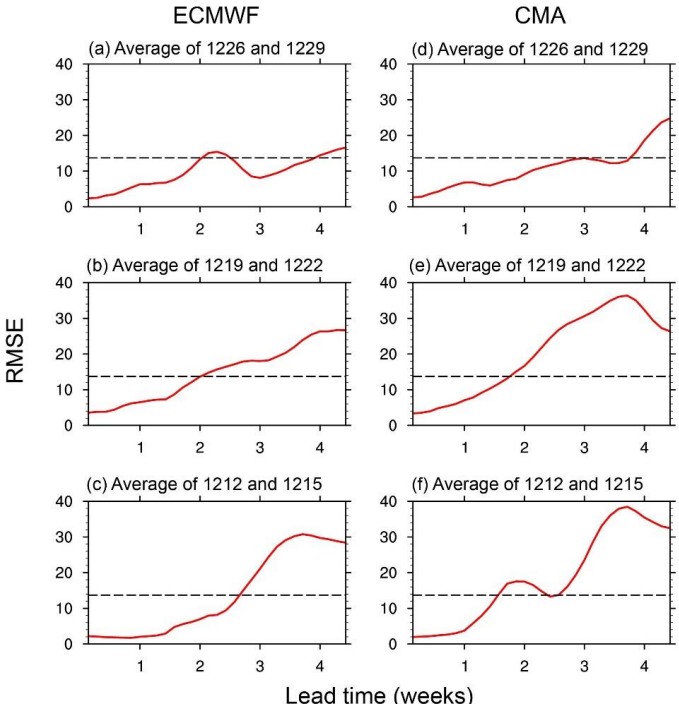

Fig. 8 Variations of RMSEs of ECMWF and CMA as a function of forecast lead time initialized
from different weeks in advance. The red solid line indicates the RMSE, and the black dashed
line indicates the average AR.

The CIS is the most distant condition to forecast the ES, and it will evolve into the ES
finally (Li et al., 2023a). Figure 9 shows the spatial distribution of the CIS of the 2021
SSW onset. The zonal westerly winds prevail mainly in the north of the Bering Strait,
the northwestern Eurasia, and the east of North America, which is resemble the spatial
structure of the zonal wind in the high latitude of the 2021 SSW onset (Fig. 7).
Meanwhile, there is a wide range of weak zonal Westley winds off Greenland in the
high latitudes, shifting to the eastly. The CIS date is the most distant initialized date,
which can be used to forecast the 2021 SSW onset. This means the CIS may play a role
of the precursor signal of the 2021 SSW onset but it needs further study.



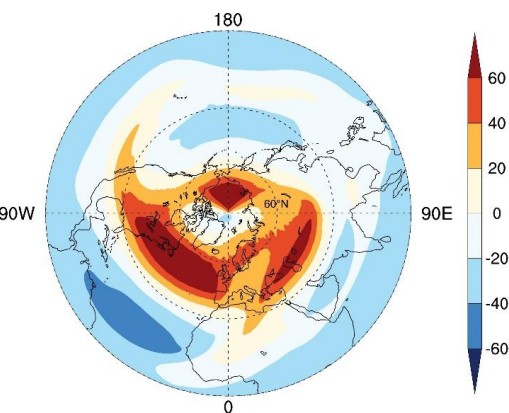

Fig. 9 As for Fig. 7, but for the CIS.

We now study the sources of forecast errors for ECMWF and CMA models in the 2021
     SSW event. Figure 10 shows the spatial structure of initial RMSE on the first day. From
     fig.10, the ECMWF and CMA models have similar spatial structures of initial errors.
     Smaller initial errors are mainly distributed north of 60°N, while larger initial errors are
     mainly distributed between 45 to 60°N. In addition, North America (60°W-150°W),
Western Europe (30°W-0°) and central Eurasia (30°E-90°E) have larger initial errors in
     both models.

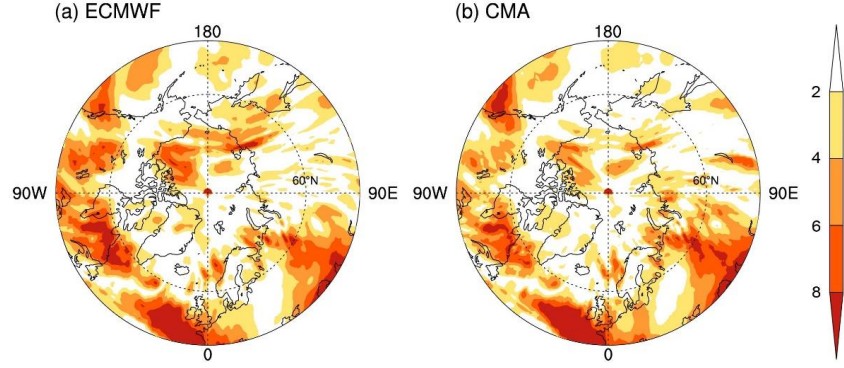

Fig. 10 Spatial structure of initial RMSE (north of 45°N) on the first day.




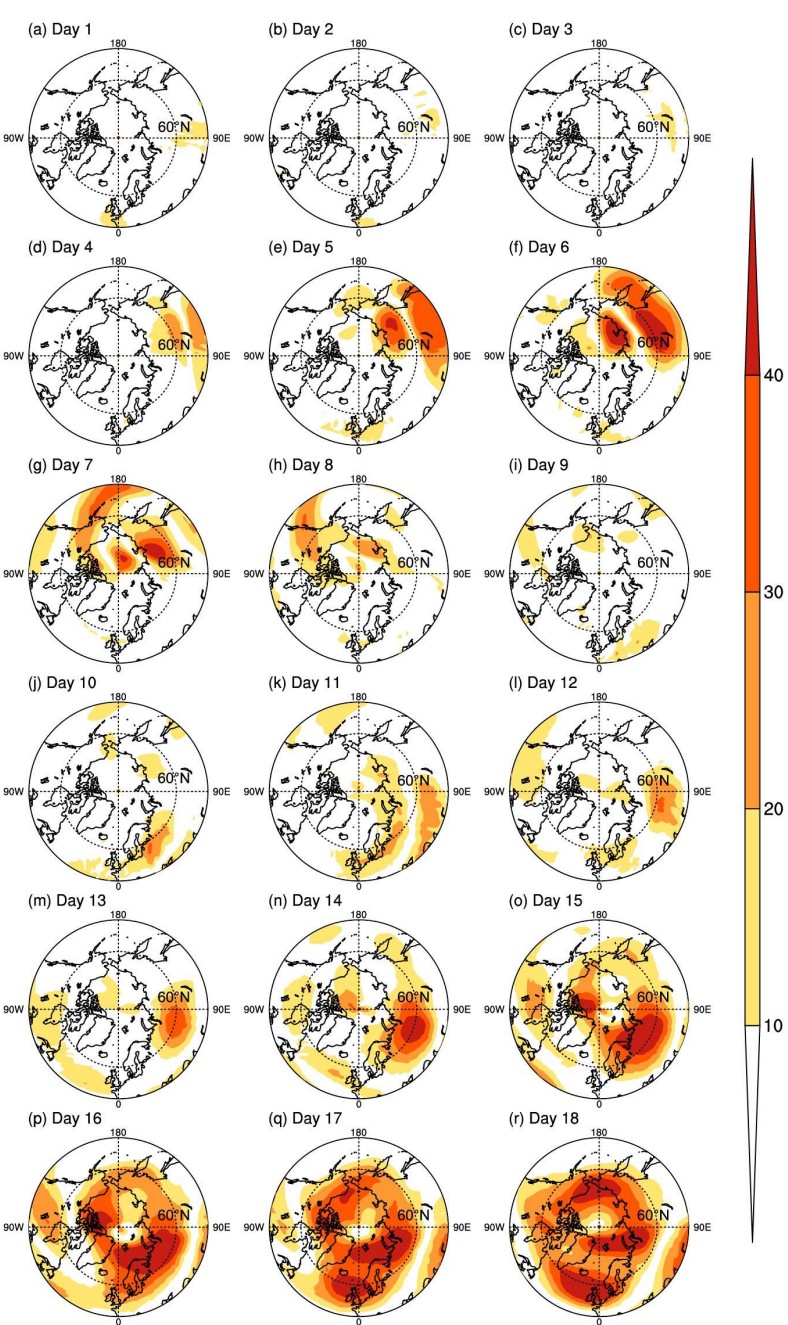

Fig. 11 As for fig.10, but for different forecast time from ECMWF model.

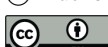



Figure 11 shows the spatial structure of the RMSEs variations with forecast time from

ECMWF model. In the first three days (Figs. 11a-c), the RMSEs are small overall. Relatively large forecast errors are mainly distributed over the middle latitudes of eastern Eurasia (near 120°E). During the following three days (Figs. 11d-f), forecast errors over Eurasia show a noticeable increase. Additionally, the coverage of forecast errors also expands. However, from the seventh to the twelfth day (Figs. 11g-l), the

forecast errors of eastern Eurasia do not continue to grow. On the contrary, they have a large decrease. Other area, especially the central Eurasia (near 90°E), show an obvious increase of forecast errors. Over the next several days, forecast errors further grow until they reach the saturation level. At this point, the forecast errors are primarily distributed between 60° and 75°N, presenting a zonal structure.

Figure 12 shows the spatial structure of the RMSEs variations from the CMA model. Same as the ECMWF model, during the early stage of the forecast lead time, the forecast errors are generally small. Only mid-high latitudes of Eurasia (near 90°E) have relatively large forecast errors. They have an obvious increase and extends eastward as the forecast time increases. During the medium stage of the forecast lead time (Figs.

12g-l), most regions of mid-high latitudes in the NH have larger forecast errors. In particular, the maximum forecast errors are mainly located over middle latitudes of western and eastern Eurasia and the mid-high latitudes of North America (Figs. 12j-l). In addition, the 60°N latitude circle is covered by these three regions. It indicates that forcast errors of the 2021 SSW event are mainly contributed from these three regions.

During the later stage of the forecast lead time, the forecast errors further grow. And they are mainly extended from 60°N to the polar circle. As the forecast time further increases, the forecast errors also reach the saturation level (Figs. 12p-r). For this moment, the predictability is lost completely, which is the same as that of ECMWF model. Differently, the forecast errors of the CMA model are larger than those of the

ECMWF model.



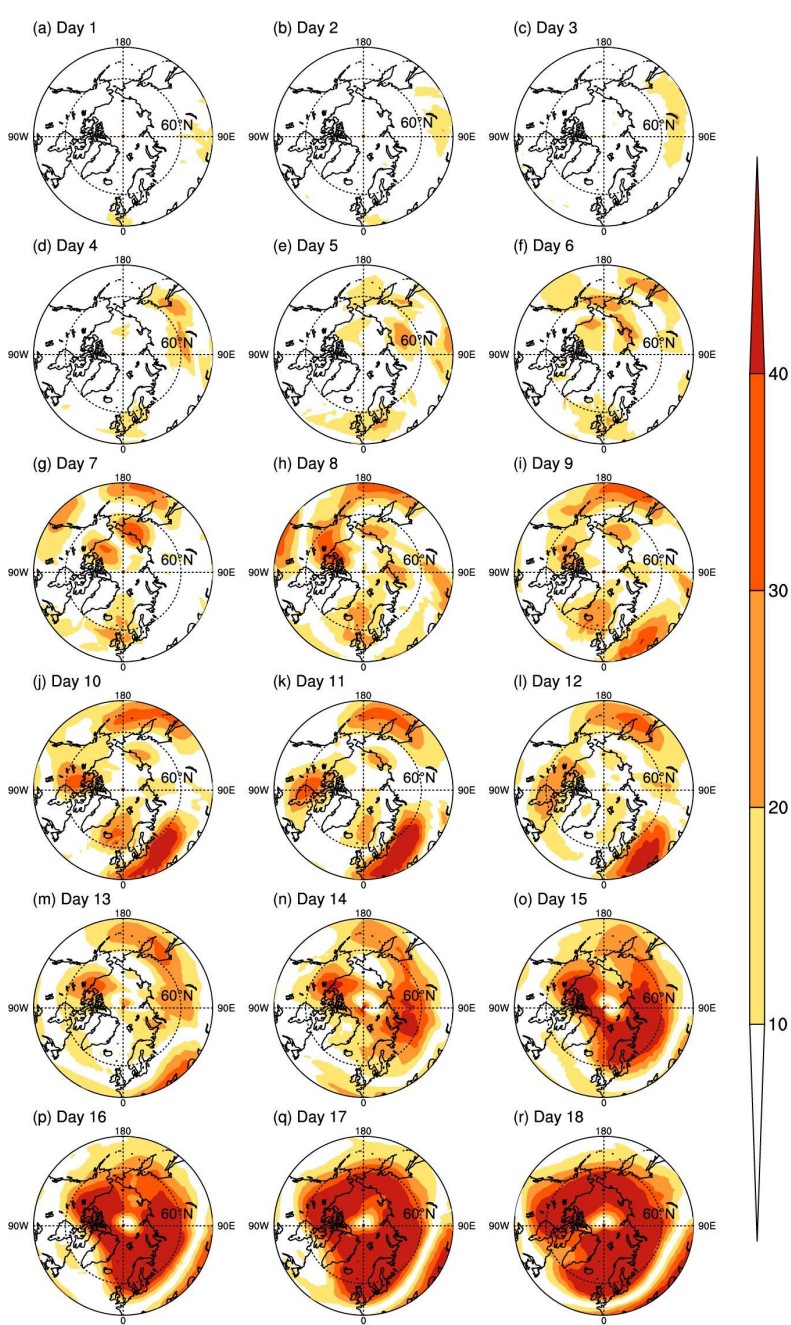

Fig. 12 As for fig.11, but for CMA model.





Generally, the growth of forecast errors in the short-term forecast is associated with two factors. Firstly, the initial error size plays a significant role, as larger initial errors tend to evolve into larger forecast errors. Secondly, the dynamics of forecast error growths is crucial. In dynamical unstable regions, forecast errors have rapid growth rate. We analyze the sources of forecast errors of the 2021 SSW event. From Fig.10, spatial

structures of initial errors are similar in ECMWF and CMA models, and larger initial errors are mainly located in the North America, Western Europe and central Eurasia. However, dynamics of forecast error growths in two models are different during the forecast time. For the ECMWF, forecast errors in central Eurasia (30°E-90°E) have grown more rapidly than other regions. Therefore, the central Eurasia can be seen the

region of unstable forecast error growths, which has large contribution of the 2021 SSW event. For the CMA, forecast errors have increased more rapidly over middle latitudes of western (0°-60°E) and eastern (120°E-180°) Eurasia and the mid-high latitudes of North America (60°W-120°W). That is, both the initial error and forecast errors during the forecast time indicate that central Eurasia (30°E-60°E) is sensitive to forecast error

growth, which limits the forecast skills of the 2021 SSW event. This suggests that the central Eurasia is a key area for the improvements of the SSW forecast.

## 4  Discussion and conclusions

In this study, a new nonlinear method, BaSIC, is employed to study the predictability of the 2021 severe SSW event using the hindcasts of the CMA and the ECMWF from

the S2S prediction database. We obtained a longer predictability of the SSW event that up to 17 days than previous linear results (no more than two weeks (Rao et al., 2021; Cho et al., 2023)). This means the S2S forecasts have the potential to predict the onset of the SSW event 17 days in advance, giving a time window for the surface weather forecast. The fact that the upper limit is longer than two weeks shows that further

improvement of the models in forecasting SSW are needed. We traced the dynamical growths of forecast error and found high altitude over central Eurasia (30°E-60°E) is the place where forecast errors originate from. This indicates the causality of the SSW in line with previous dynamical studies and have great implications for future investment of model improvement.



We note the upper limit of the predictability can depend on the model. When the initialized date is 19 December of 2020, the RMSE of the ECMWF model still can exceed the AR roughly at the 2021 SSW onset, indicating that the upper predictability limit is 17 days. However, for the CMA model, the upper predictability limit is less than two weeks. The reason why two models get two different upper predictability limits is

the presence of model errors in two models. That is, the calculated results are actually the practical predictability limits, not the intrinsic predictability limits. Even so, the practical predictability obtained by the state-of-the-art model is closer to the intrinsic predictability. In addition, the practical predictability limit (17 days) of the 2021 SSW event is higher than that in previous studies (no longer than two weeks). From our

perspectives, the main reason is the choice of predictability methods. The BaSIC method employed in this work takes the nonlinearity of error growths in the stratosphere, while some previous studies use the linear methods, such as the anomaly correlation coefficient or signal to noise ratio. Furthermore, this work also investigates the geographical sources of forecast error growths for the 2021 SSW event. By exploring

the distribution of the initial error size and the dynamical growths of forecast error, the central Eurasia (30°E-60°E) is sensitive to forecast error growths in both models, indicating this geographical region has larger contributions to forecast errors of the 2021 SSW event. For sensitive regions of forecast error growths, it is generally recognized that adding more observation stations and using more advanced data

assimilation techniques could effectively generate more accurate initial analysis conditions, thereby improving the forecast skills. This work gives the sensitive regions of forecast errors for the 2021 SSW event, and it is worthy verifying it in further work.

## Appendix:

1. Attractor radius and global attractor radius

Li et al. (2018) introduced the attractor radius (AR) and global attractor radius (GAR) to characterize the dynamics of chaotic systems. Based on the two metrics, the average predictability of chaotic systems can be quantified. However, the two metrics fail to estimate the local predictability of given events, especially the extreme events. To study

the local predictability of specific states, the BaSIC method, derivative of AR and global AR, was developed to investigate the local predictability of specific states (Li et al.,



2023b). Before the introductions of the BaSIC method, we will firstly present the information of the AR and GAR.

For a compact attractor $\mathcal{A}$, the distance between a specific state $\boldsymbol{x}_i$ and all other states on the attractor can be expressed as follows:

$$R_L(\boldsymbol{x}_i) = \sqrt{E(\| \boldsymbol{x}_i - \boldsymbol{x} \|^2)}, \ \boldsymbol{x}_i \ \text{and} \ \ \in \mathcal{A}, \tag{A1}$$

where E means the expectation, and $\| \ \|$ represents the $L_2$ norm of the vector. The local distance $\boldsymbol{R}_L$ is called the local AR and it varies with the selected local state $\boldsymbol{x}_i$. Specifically, if the local state $\boldsymbol{x}_i$ is just the average state $\boldsymbol{x}_E$ of the compact attractor $\mathcal{A}$, the local distance can be denoted as follows:

$$R_E = \sqrt{E(\| \boldsymbol{x}_E - \boldsymbol{x} \|^2)}, \text{and} \ \ \in \mathcal{A}, \tag{A2}$$

where $\boldsymbol{R}_E$ is the AR. Although $\boldsymbol{x}_E$ denotes the average state of the attractor $\mathcal{A}$, it does not always fall on the attractor.

Averaging all LARs of the states $\boldsymbol{x}$ on the compact attractor $\mathcal{A}$, the GAR of the compact attractor $\mathcal{A}$ can be obtained, and can be expressed as follows:

$$R_G = \sqrt{E(\boldsymbol{R}_L{}^2)} = \sqrt{E(\| \boldsymbol{x} - \boldsymbol{y} \|^2)}, \ \boldsymbol{x} \ \text{and} \ \boldsymbol{y} \in \mathcal{A}. \tag{A3}$$

Actually, $\boldsymbol{R}_G$ indicates the average distance between any two states on the compact attractor $\mathcal{A}$. And there is a constant relationship between the AR and GAR. That is

$$R_G = \sqrt{2}R_E. \tag{A4}$$

2. Quantifying the local predictability

This study mainly addresses the local predictability of extreme events. Therefore, we will introduce how to estimate the local predictability.

For an n-dimensional dynamical system, a perturbed state $\widehat{\boldsymbol{x}}_0$ can be derived by superimposing the initial error $\boldsymbol{\delta}_0$ on the initial state $\boldsymbol{x}_0$. It can be expressed as follows:

$$\widehat{\boldsymbol{x}}_0 = \boldsymbol{x}_0 + \boldsymbol{\delta}_0. \tag{A5}$$

Then the dynamic trajectory of the two local states $\boldsymbol{x}_0$ and $\widehat{\boldsymbol{x}}_0$ varying with time can be expressed as $\boldsymbol{x}(t)$ and $\widehat{\boldsymbol{x}}(t)$. Meanwhile, the root-mean-square error (RMSE) is a function of forecast time and can be denoted by





$$e(\boldsymbol{x}_0, \boldsymbol{\delta}_0, t) = \parallel \hat{\boldsymbol{x}}(t) - \boldsymbol{x}(t) \parallel. \tag{A6}$$

The local ensemble average of the RMSEs where a large number of initial errors are superimposed on the initial state $\boldsymbol{x}_0$ is defined as:

$$\bar{e}(\boldsymbol{x}_0, \boldsymbol{\delta}_0, t) = \sqrt{\langle e^2(\boldsymbol{x}_0, \boldsymbol{\delta}_0, t) \rangle_N}, \ \boldsymbol{x}_0 \in \mathcal{A}, \tag{A7}$$

where N denotes the number of initial errors, and $\langle . \rangle_N$ means the ensemble mean of $N$

samples. When $\bar{e}(\boldsymbol{x}_0, \boldsymbol{\delta}_0, t)$ reaches the AR, it indicates the local predictability limit (LPL) of the state $\boldsymbol{x}_0$ (Li et al., 2018), which can be described by

$$T_{LPL,x_0} = t_{ar} - t_0, \tag{A8}$$

and $t_{ar}$ represents the time when $\bar{e}(\boldsymbol{x}_0, \boldsymbol{\delta}_0, t)$ reaches the AR. Based on equation (8), the practical accurate forecast time starts from the state $\boldsymbol{x}_0$ can be accurately

obtained.

**Data availability:**

The ECMWF and CMA reforecasts data are freely available at https://apps.ecmwf.int/datasets/data/s2s-reforecasts-daily-averaged-

ecmf/levtype=sfc/type=cf/. The ERA5 reanalysis dataset are freely available from the Climate Data Store https://cds.climate.copernicus.eu/datasets/reanalysis-era5-pressure-levels?

**Author contribution:**

The study was designed by XZ and XL, and they contributed to the interpretation of the results and the writing of the manuscript. GZ produced the figures and prepared the manuscript, assisted by YL and QC for review and editing. All the authors contributed to editing and reviewing.

**Competing interests:**

The authors declare that they have no conflict of interest.



## Acknowledgements:

This work was jointly supported by the National Natural Science Foundation of China
(U20A2097, 42275059, 42175042), and Natural Science Foundation of Sichuan
Province (2024NSFTD0017, 2023NSFSC0246). We gratefully acknowledge the
European Centre for Medium-Range Weather Forecasts (ECMWF) for providing
subseasonal to seasonal dataset and the ERA5 reanalysis dataset.

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
