# Peer review of "Quantifying the local predictability of the 2021 sudden stratospheric warming event using a novel nonlinear method"

_EGUsphere, 2024_

## Author Comment (AC1)

**Responses to Reviewers**

**Submission ID:** EGUSPHERE-2024-2574

**Title:** Quantifying the local predictability of the 2021 sudden stratospheric warming event using a novel nonlinear method

**Author(s)**: Guiping Zhang, Xuan Li, Yang Li, Quanliang Chen, and Xin Zhou

**January 17, 2025**

**Responses to Reviewer 2**

**II-A: Summary**

*The manuscript investigates the predictability limit of the 2021 Northern Hemisphere Sudden Stratospheric Warming (SSW) event using a fairly novel technique known as BaSIC (BAckward Searching for the Initial Condition). Using ERA5 as verification, the authors choose two subseasonal-to-seasonal (S2S) model forecasts in which to show how far in advance the dynamical models could predict the SSW event. As expected, the ensemble-mean forecasts have the best skill (space and time) under 2 weeks before the start date of the SSW event, agreeing with previous works. However, using the BaSIC method on zonal wind conditions, the authors find that predictability may be extended to 14-17 days before the event. Then, the authors examine error growth in stratospheric fields to show that the largest errors in the stratosphere for this event exist over the middle to high latitudes of Eurasia and North America, suggesting that these regions may be the biggest contributors to the errors in the forecast models' performance for the 2021 SSW.*

**Response:** Thank the reviewer for sparing time to go through the manuscript and pointing out many important comments.

**II-B: Overall Opinion**

*The manuscript presents an interesting way of trying to understand how to examine predictability limits for extreme events like major SSWs. However, I have several concerns about the paper that make it not ready for publication at this time. These concerns are mainly twofold: (1) the BaSIC methodology is inadequately described and, with all of the acronyms used, is very confusing to follow, and (2) the paper lacks actual scientific advancement in identifying the source of errors, particularly when it only focuses on stratospheric fields. The writing is generally good but could also use some more proofreading and refinement. At this time, I think that the paper requires significantly more work before it can be published. Therefore, I am unfortunately recommending that it be rejected with the opportunity for resubmission later.*

**Response:** Thank the reviewer for sparing time to go through the manuscript, highlighting very important issues and providing helpful comments and valuable suggestions to improve the manuscript. According to the reviewer's suggestions, we

have revised the manuscript seriously and carefully. More details and point-to-point responses to the reviewer's comments are listed as follows.

**II-C: Major Concern**

*1.* **Poor Description of BaSIC.** *While the authors have referenced other papers that use the BaSIC technique, there is a lack of clarity on several terms, equations, and their use throughout the paper. For example, I am very uncertain what CIS and ES refer to (Lines 161 and 162 and elsewhere). I am also unclear what it means that a random state vector* ***x0*** *"loses its predictability" (Line 167) - e.g., what is the measure for "predictability?" Much more clarity in the methods section needs to be applied in order for just basic understanding of the method, which can then be used for further critiquing its use in the paper.*

**Response:** Thanks for the reviewer pointing out this issue, and apology for poor description of BaSIC method. The BaSIC technique used in our study aims to quantify the predictability limit of specific events from the perspective of forecast error growths. The criterion on how to estimate predictability limit includes three steps. Firstly, we should determine a specific or target state, whose predictability limit is to be quantified. Because the predictability limit of target state estimated by the BaSIC method is based on the perspective of nonlinear growth of forecast errors. Secondly, we need to investigate the nonlinear growth of forecast errors. Specifically, the growth of initial forecast errors from different initialization date should be analyzed. Thirdly, after the analysis of initial forecast error growths, corresponding initial state (CIS) should be determined. Determining the corresponding initial state should follow a criterion. If the forecast error of an initial state prior to this target state grows with time, and exceeds the threshold value at the time of the target state, then this initial state is the CIS which we need to find. After the determination of the CIS, the timespan between the corresponding initial state and the target state is defined as the predictability of the target state. It should be noted that the threshold value employed is the attractor radius, which is the standard deviation of a variable in a long time series. The above mentioned steps are how to estimate predictability limit of a specific state.

To make readers more understandings on the BaSIC technique, we have rewritten the

introduction of this method. More details can be found in the revised manuscript (Lines 169-197 and 614-710).

In addition, the ES and CIS refer to extreme state and corresponding initial state, respectively. The extreme state and ES have been replaced with target state and TS in the revised manuscript, respectively.

Apology for the poor description of predictability. Clarifications on "loses its predictability" have been added from lines 687 to 701. Predictability is closely related to the growth of the initial error. Slight difference of two states ($x_0$ and $x'_0$) on initially nearby trajectories in phase space grows over time. Owing to the chaotic nature of nonlinear dynamical systems, the slight difference will evolve to exceed a certain threshold value, then predictability of the state $x_0$ can be considered to be lost.

Detailed procedure on how to measure the practical predictability of the 2021 SSW event can be found in the revised manuscript from lines 169 to 197.

*2. Does this method really show longer predictability windows? The authors make a key point of highlighting that, while the dynamical models in the aggregate (key term) have poor prediction skill at leads longer than 2 weeks, the BaSIC technique highlights that the 2021 major SSW event had a predictability limit of 14-17 days. I have a couple of issues with this statement. First, the authors are comparing the skill for ONE event against the skill of models for MANY events. For any extreme event phenomena (heavy rainfall, heatwaves, etc.), one can find evidence that even a model can perform better than expected for MULTI-MODEL means for ONE event versus a collection of them. This is why we have aggregate statistics for model performance. Scientific interest certainly lies in outlier events (e.g., events in which the model did exceptionally well or poorly forecasting at long leads), but the comparison here of an aggregate of 14 days predictability limit vs. 14-17 days for the BaSIC method isn't the right one, in my opinion. Moreover, is a gain of maybe 3 days really useful or even significant for this event? Overall, I find this to be a significant weakness of the authors' main argument for this manuscript.*

**Response:** Thanks for the good comments from the reviewer. The BaSIC method employed in the study reveals longer predictability windows than operational forecast skills of two S2S models. It should be noted that the BaSIC method focuses on estimating the practical predictability of specific events. And the practical predictability

is different from the forecast skills. Actually, the practical predictability is the upper limit of forecast skills with the presence of both the initial errors and model errors. In this study, we firstly evaluated the forecast skills and found the ECMWF and CMA models show relatively good performances within two weeks. However, the practical predictability limit is higher than the forecast skills. Then, how long is the practical predictability limit? We are more concerned with this issue. Therefore, the key point of this work is to quantify the practical predictability limit of the 2021 major SSW event, and we use the BaSIC method to address this issue. The result showed that the 2021 major SSW event had a predictability limit of 17 days using the two single models. We have rewritten the BaSIC method to make potential readers more understand it (Lines 169-197 and 614-710).

Apology for poor descriptions to lead misunderstandings. In practice, we haven't compared the skill for one event against the skill of models for many events. We only focus on the 2021 SSW event, and are not concerned with other SSW events. It should be noted that we have evaluated the average forecast skills of the zonal wind (north of 60°N, 10-hPa) within 15 days by ECMWF and CMA centers from December 2020 to January 2021 (Fig. 3). However, this evaluation of average forecast skills in winter is independent from that of the forecast skills of the 2021 SSW. In addition, we also have not carried out the comparison of forecast skills between one model and multi-model means. In this work, we only use two S2S models to investigate the practical predictability of the 2021 SSW event. During the investigation, we preformed analysis using the reforecast data from the respective model. And we have not carried out the analysis of multi-model means. Since the multi-model means are not needed in evaluation of average forecast skills or quantifying the upper limit of practical predictability. How to obtain the practical predictability limit (17 days) of the 2021 SSW event, and corresponding analysis can be found from lines 412 to 455 of the revised manuscript.

For synoptic-scale events, the practical predictability limit is widely recognized within two weeks. However, the SSWs are subseasonal to seasonal events. After applying the BaSIC method to the 2021 SSW event, we obtained a practical predictability limit of

17 days, further demonstrating that the SSW event has a longer predictability window. Besides, we also have not performed the comparison of an aggregate of 14 days predictability limit vs. 14-17 days for the BaSIC method. In fact, the time lengths of two S2S reforecasts are 46 and 60 days for ECMWF and CMA models, respectively. We just showed the reforecasts of 14 days, since the reforecasts of longer time have little forecast skills. Anyway, the evaluation of forecast skills in 14 days is independent from the upper limit of practical predictability of 17 days calculated by the BaSIC method.

We have added more clarifications to make potential readers more understand our work in the revised manuscript (Lines 412-455).

*3. Source of Errors is too limited and does not offer significant scientific advancement. In using their novel technique, the authors also discuss how they are able to find sources of errors for the forecasts which subsequently can result in poorer predictability. However, the authors only look at stratospheric fields, motivated by some literature cited in the Introduction. While certainly there can errors in the evolution of these fields (or maybe even initialization errors), another major source for changes in the polar stratospheric flow fields is wave driving from the troposphere. Models can have significant errors (and/or biases) in wave driving, which subsequently cascade into stratospheric circulation errors. For example, Schwartz et al. (2022) showed that biases in stationary waves within several subseasonal prediction models significantly impact the upward propagation of waves into the stratosphere. This paper does not even consider these errors for this major SSW. I also do not see suggestions for ways to improve even the errors shown in the paper within the Discussion section. These types of analyses and suggestions are where the scientific advancements could be made with this paper and thus make it a useful publication. As such, I recommend that the authors rework their "sources of error" section of the paper to account for tropospheric fields and wave driving and make suggestions for improvement.*

**Response:** Thanks for the reviewer's suggestion. We are sorry for the error source section to lead some misunderstandings. In fact, we would like to reveal the sensitive areas of forecast error growths based on the ERA5 reanalysis and S2S reforecasts. It is mainly because forecast errors over sensitive areas grow more rapidly than other areas, limiting the overall forecast skills. Investigating the source of errors from external

conditions is not our intention in this work. We agree that it is of great scientific interest to study sources of errors, and we will work on the source of errors from external conditions in further work. To eliminate the potential misunderstandings, we have added subheadings in this section. And the subheading of this section is "Evolution of sensitive forecast error growths".

We agree that tropospheric wave activities have influenced the stratospheric flows, and errors in tropospheric wave driving will cascade into stratospheric circulation errors. However, our study is not involved with the external sources of the practical predictability of 2021 SSW event. The aim of this work is to directly estimate its practical predictability limit based on the reanalysis data and the reforecast data. The reason why we use stratospheric data is that the observational data contained all the dynamical information from the external factors. Researchers have successfully obtained predictability limits of climate and weather events based on the observational data (Ding et al., 2010; Li and Ding, 2013; Ding et al., 2016). Some references are listed below. We hold the same opinion as theirs. In this study, we think that the zonal wind data at 10-hPa contained the dynamical information both from stratospheric and tropospheric wave activities. Therefore, to obtain the practical predictability limit of the 2021 SSW event, we just directly need to analyze the reanalysis data and reforecast data.

Anyway, investigating the tropospheric error sources of practical predictability limit of the 2021 SSW event have important scientific significances. And we will carry out it in further work.

We have added more explanations in the revised manuscript from lines 354 to 368, and from lines 501 to 508.

Ding, R., Li, J. and Seo, K.-H. Predictability of the Madden–Julian oscillation estimated using observational data. Monthly Weather Review 138, 1004-1013, https://doi.org/doi.org/10.1175/2009MWR3082.1, 2010.

Li, J. and Ding, R. Temporal-spatial distribution of the predictability limit of monthly

sea surface temperature in the global oceans. International Journal of Climatology 33, https://doi.org/doi.org/10.1002/joc.3562, 2013.

Ding, R., Li, J., Zheng, F., Feng, J. and Liu, D. Estimating the limit of decadal-scale climate predictability using observational data. Climate Dynamics 46, 1563-1580, https://doi.org/doi.org/10.1007/s00382-015-2662-6, 2016.

**II-D: More Minor/Specific Comments**

*1. **Line 10.** Change the start of the sentence to "Sudden stratospheric warmings (SSWs) are..." Same with **Line 32**.*

**Response:** Revised (Lines 11 and 33).

*2. Line 48. Change "tropospheric fluctuations" to "tropospheric waves."*

**Response:** We have revised it (Line 49).

*3. Figure 2 (and others). The pressure level at which these fields are plotted is not indicated in the caption or accompanying text. This is very important to indicate and should be labeled.*

**Response:** Our analyses are all based on the pressure level of 10-hPa. We have labeled in the revised manuscript (Lines 266-268).

*4. Figure 3. There is no indication the months/season to which these errors refer. Please be more specific in the figure caption and text.*

**Response:** Figure 3 presents the average daily forecast errors from December 2020 to January 2021. We have added specific clarifications in the figure caption in the revised manuscript (Lines 281-283).

*5. Figure 4 and accompanying text. There is no discussion of how different ensemble members perform (note, for example, there are some ECMWF ensemble members show a major SSW at 2 week leads). This is important, as the authors only compare their method to ensemble-mean statistics.*

**Response:** We have redrawn this figure and added more discussions of how different ensemble members perform in the revised manuscript (Lines 301-329).

*6. Lines 269-270. I don't know what "lost their forecast skills" means exactly. Is there a quantitative measure for loss of skill?*

**Response:** Apology for unclear description. We intended to demonstrate the forecast skills decreased so much that the accuracy of forecasts was very low. To eliminate the misunderstandings, we have changed it to " showed lower forecast skills" in the revised manuscript (Line 342). Generally, we qualitatively describe the forecast skills from the forecast errors. If the forecast errors are large, the forecast skills are low, and vice versa.

*7. Figure 5. There is no indication of what field is plotted or what the units are.*

**Response:** At 10-hPa, and the unit is m/s. We have illustrated this in the revised manuscript (Lines 345-347).

*8. Appendix. I do not know what the script "A" represents in Equations A1, A2, and A3.*

**Response:** $\mathcal{A}$ is the calligraphic style of script "A" in Equations A1, A2, and A3, and it represents the compact attractor (Lines 641).

*9. Data Availability. There is no indication that the authors are making their code and/or datasets publicly accessible. Please consider adding a Zenodo and/or Github repository for this.*

**Response:** We have created an Zenodo link and put it in Code and data Availability (Lines 717-719).

---

## Author Comment (AC2)

**Responses to Reviewers**

**Submission ID:** EGUSPHERE-2024-2574

**Title:** Quantifying the local predictability of the 2021 sudden stratospheric warming event using a novel nonlinear method

**Author(s)**: Guiping Zhang, Xuan Li, Yang Li, Quanliang Chen, and Xin Zhou

**January 17, 2025**

**Responses to Reviewer 1**

**I-A: General comment**

*The authors apply a non-linear analytical assessment of the stratospheric state to assess the local predictability leading up to the 2012 Sudden Stratospheric Warming. The study provides an additional metric for assessing predictability alongside more conventional (linear) approaches and is therefore of merit for publication. However, I feel the manuscript needs substantial modification to enable readers to understand the applied method, the analysis carried out, and the comparison with other dynamical studies (see comments below).*

**Response:** Thank the reviewer for sparing time to go through the manuscript, highlighting very important issues and providing helpful comments and valuable suggestions to improve the manuscript. According to the reviewer's suggestions, we have revised the manuscript seriously and carefully. More details and point-to-point responses to the reviewer's comments are listed as follows.

**I-B: Specific comments**

*1. The methodology needs a clearer explanation – the authors have referenced previous papers, however more detail on the actual calculations and steps performed on the datasets within this study are needed for a reader to follow. As a minor example, line 181 refers to equations 9 and 10 however these are not found in the manuscript.*

**Response:** Thanks for the good suggestions from the reviewer. We have rewritten the methodology and add more descriptions on the actual calculations and steps performed on the ERA5 reanalysis and S2S reforecast datasets in the reversed manuscript (Lines 169-197).

The BaSIC technique used in our study aims to quantify the practical predictability limit of specific events from the perspective of forecast error growths. The criterion on how to estimate predictability limit includes three steps.

Step 1. We should determine a specific or target state, whose predictability limit is to be quantified. Because the predictability limit of target state estimated by the BaSIC method is based on the perspective of nonlinear growth of forecast errors.

Step 2. We need to investigate the nonlinear growth of forecast errors. Specifically, the

growth of initial forecast errors from different initialization date should be analyzed.

Step 3. After the analysis of initial forecast error growths, corresponding initial state (CIS) should be determined. Determining the corresponding initial state should follow a criterion. If the forecast error of an initial state prior to this target state grows with time and exceeds the threshold value at the time of the target state, then this initial state is the CIS which we need to find. After the determination of the CIS, the timespan between the corresponding initial state and the target state is defined as the predictability of the target state. It should be noted that the threshold value employed is the attractor radius, which is the standard deviation of a variable in a long time series. Above descriptions are the general procedure to quantify the practical predictability limit of specific events based on the BaSIC method. More details on how to apply the BaSIC method to practical predictability limit of the 2021 SSW event can be found in from lines 174 to 190.

In addition, Equations 9 and 10 now refer to equations A7 and A8 in the revised manuscript (Lines 703-705).

*2. Apologies if I have misunderstood, but it appears the analysis performed here relates only to the zonal wind field at 10hPa, and is therefore missing the influence (and error growth) within the upper troposphere and lower stratosphere (which Cho et al 2021 suggest are important for this event)? The authors discuss the regional error growth within the forecast models in the stratosphere, however this is likely to be driven by the tropospheric wave activity, and will thus affect the resulting predictability limits?*

**Response:** Thanks for the reviewer's suggestion. It is true that we only analyzed the zonal wind data at 10-hPa. And we agreed with the point from Cho et al. (2021). And the upper troposphere and lower stratosphere can modulate the predictability of 2021 SSW event. In fact, our study is not conflicted with that of Cho et al. (2021). Cho et al. (2021) aimed to find the important source of predictability of 2021 SSW event, and then determined its predictability. However, our study is not involved with the sources of the predictability of 2021 SSW event, and we directly estimated its predictability limit based on the reanalysis data and the reforecast data. Therefore, the research approach of our study is different from that of Cho et al. (2021).

From some previous papers, the observational data is directly analyzed, and then predictability limits of climate and weather events are quantified (Ding et al., 2010; Li and Ding, 2013; Ding et al., 2016. These studies don't need to find the sources of predictability, because they deemed that the observational data contained all the dynamical information from the external factors. We hold the same opinion as theirs. In this study, we think that the zonal wind data at 10-hPa contained the dynamical information both from stratospheric and tropospheric wave activities. Therefore, to obtain the practical predictability limit of the 2021 SSW event, we just directly need to analyze the reanalysis data and reforecast data.

We have added more explanations in the revised manuscript from lines 354 to 368, and from lines 501 to 508.

Ding, R., Li, J. and Seo, K.-H. Predictability of the Madden–Julian oscillation estimated using observational data. Monthly Weather Review 138, 1004-1013, https://doi.org/doi.org/10.1175/2009MWR3082.1, 2010.

Li, J. and Ding, R. Temporal-spatial distribution of the predictability limit of monthly sea surface temperature in the global oceans. International Journal of Climatology 33, https://doi.org/doi.org/10.1002/joc.3562, 2013.

Ding, R., Li, J., Zheng, F., Feng, J. and Liu, D. Estimating the limit of decadal-scale climate predictability using observational data. Climate Dynamics 46, 1563-1580, https://doi.org/doi.org/10.1007/s00382-015-2662-6, 2016.

*3. Once again, apologies if I have misunderstood, but a key assumption within this methodology is that any state preceding the identified initial condition cannot, by definition, predict the event state? Given the importance of tropospheric conditions driving the stratospheric flow, by only focusing on the 10hPa winds, surely multiple tropospheric states could precede the event (at lead times earlier than the identified initial condition) and would not be captured within this analysis? A practical example of this is that you may find one member predicts the event at a very long lead time (earlier than those utilised here), possibly by chance, but this still represents a physical state leading to the SSW.*

**Response:** Thanks for the good comment from the reviewer. This methodology firstly needs to find a corresponding initial condition, which is prior to the target condition. In this study, the target condition is the reversal of westerly zonal-mean zonal wind to easterly at 10-hPa (that is the onset of 2021 major SSW event). After determining the corresponding initial condition, the timespan between the corresponding initial condition and the target condition is defined as the predictability limit of the 2021 SSW event. A criterion on how to determine the corresponding initial condition is that the corresponding initial condition is the most distant condition to predict the target condition. That is, any state between the corresponding initial condition and the target condition can predict the target condition. This clarification can be found in from lines 169 to 197 of the revised manuscript.

We think that the influence of both the stratospheric and tropospheric process will reflect in the reanalysis and reforecast data. Therefore, directly analyzing the zonal wind at 10-hPa can estimate the predictability limit of the onset of the 2021 SSW event, whose predictability is affected by both the stratospheric and tropospheric wave activities.

In this study, we are more concerned with the ensemble mean of all members of reforecasts, not the single member. Because the ensemble mean result will eliminate the effects on predictability from some bad members. In addition, a single good member may have a certain degree of randomness. That is this member performs well in this forecast, but bad in other forecasts. Therefore, we don't analyze the single member, but use the ensemble mean result to study the predictability.

**I-C: Minor comments**

*1. Suggest adding "practical" to the article title (i.e. "quantifying the practical local predictability…"*

**Response:** We have added "Practical" to the article title.

*2. Please can sub-headings be used to separate out the different sections of the results?*

**Response:** We have divided the results into three parts in the revised manuscript.

*3. How does this approach compare or with other statistical approaches e.g. Finkel et al 2023 (Revealing the Statistics of Extreme Events Hidden in Short Weather Forecast Data)?*

**Response:** Thanks for the comment. Finkel et al. (2023) presented an effective approach to reveal statistics of extreme events. This method can extract climatological information from these short weather simulations, and then characterize sudden stratospheric warming (SSW) events with multi-centennial return times. In addition, the estimates of the frequencies and seasonal distributions of SSW can be also obtained. The BaSIC method used in this work investigates the practical predictability of the 2021 SSW event from the nonlinear perspective. This method focuses on the dynamics of forecast errors to quantitatively estimate the upper practical predictability limit of the 2021 SSW event. Both two methods are effective tools to study the predictability of SSW events.

We have added the discussion of the work of Finkel et al (2023) in the introduction section (Lines 89-95).

*4. If understood correctly, the analysis depends upon the ensemble mean RMSE; can the authors add discussion regarding the regional growth of the ensemble spread, which is also important for understanding predictability barriers (e.g. Sanchez 2020: Linking rapid forecast error growth to diabatic processes)?*

**Response:** We have added more analysis of the regional growth of the ensemble spread (Fig. 5, Lines 286-299; Figs. 11 and 12, Lines 472-497). It is showed that the spread of reforecasts in both ECMWF and CMA models is lower than the root-mean-square error (RMSE). And the spread of ECMWF is higher than that of CMA. The differences in spread arise from different ensemble numbers and different initialization schemes in two numerical centers. Generally, the spread is an indicator of forecast skills. The closer the Spread is to the RMSE, the higher the forecast skills.

*5. It is also useful to note that new dynamical methodologies demonstrating causality are now available and have been used for SSWs (Kent et al 2023: Identifying Perturbations That Tipped the Stratosphere Into a Sudden Warming During January 2013)*

**Response:** We have added the discussion of the work of Finkel et al (2023) in the introduction section (Lines 82-88).

*6. There are several acronyms which need defining (e.g. WN1, ES, CIS)*

**Response:** WN1 refers to zonal-wavenumber 1. And the ES and CIS refer to extreme state and corresponding initial state, respectively. The extreme state and ES have been replaced with target state and TS in the revised manuscript, respectively.

*7. In the abstract the authors state that the surface response is around two weeks following SSWs (Line 11), however the surface impact can be 30-60 days for the downward coupling to influence the troposphere.*

**Response:** We have fixed it in the revised manuscript (Lines 12-13).

*8. A few spelling errors also need addressing (e.g. "middel")*

**Response:** We have revised it (Line 403).